# Advancement and Potential Applications of Epididymal Organoids

**DOI:** 10.3390/biom14081026

**Published:** 2024-08-17

**Authors:** Junyu Nie, Hao Chen, Xiuling Zhao

**Affiliations:** Institute of Reproductive Medicine, Medical School, Nantong University, Nantong 226019, China; njy@ntu.edu.cn (J.N.);

**Keywords:** epididymal organoid, male infertility, epididymal epithelial, basal cell

## Abstract

The epididymis, a key reproductive organ, is crucial for sperm concentration, maturation, and storage. Despite a comprehensive understanding of many of its functions, several aspects of the complex processes within the epididymis remain obscure. Dysfunction in this organ is intricately connected to the formation of the microenvironment, disruptions in sperm maturation, and the progression of male infertility. Thus, elucidating the functional mechanisms of the epididymal epithelium is imperative. Given the variety of cell types present within the epididymal epithelium, utilizing a three-dimensional (3D) in vitro model provides a holistic and practical framework for exploring the multifaceted roles of the epididymis. Organoid cell culture, involving the co-cultivation of pluripotent or adult stem cells with growth factors on artificial matrix scaffolds, effectively recreates the in vivo cell growth microenvironment, thereby offering a promising avenue for studying the epididymis. The field of epididymal organoids is relatively new, with few studies focusing on their formation and even fewer detailing the generation of organoids that exhibit epididymis-specific structures and functions. Ongoing challenges in both clinical applications and mechanistic studies underscore the importance of this research. This review summarizes the established methodologies for inducing the in vitro cultivation of epididymal cells, outlines the various approaches for the development of epididymal organoids, and explores their potential applications in the field of male reproductive biology.

## 1. Introduction

The epididymis is characterized by an intricately coiled and convoluted tubular architecture, primarily segmented into caput, corpus, and cauda [1,2,3,4]. In rodents, an initial segment (IS) is located between the efferent ducts and the caput [5]. The caput, the initial segment, interfaces with the testicular efferent ducts, while the cauda connects to the vas deferens [6]. The length of the epididymis varies among species, exemplified by a human unfolded length of approximately 6 m, where sperm migrate from the caput to the cauda within a period of 1–2 weeks [6,7]. Each of these distinct regions possesses unique anatomical and physiological characteristics. The epithelial cells comprising the epididymal duct wall are its primary cellular component, forming a pseudostratified epithelium composed of multiple cell types (see Figure 1) [8,9,10]. These cell types include principal cells, basal cells, clear cells, apical cells, and narrow cells [6,11]. Principal cells are the predominant type, constituting 60–80% of the epithelium throughout the tubule [6,12], while basal cells make up 6–30% [13,14]. Narrow and apical cells are predominantly found in the initial segment of the epididymis, whereas other cell types are distributed throughout the entire epididymal tissue [8,11,15,16,17]. The detailed functions and protein markers of the epithelial cells are listed in Table 1. In addition, a small number of non-epithelial cells, such as macrophages/monocytes, mononuclear phagocytes, and T lymphocytes, are shown [18,19,20,21,22,23].

The in vitro culture of the epididymal epithelium is frequently employed to study the role of the epididymis in sperm maturation and the associated molecular mechanisms. In 1986, studies reported that epididymal epithelial cells were successfully isolated from adult rats and exhibited polarized characteristics only when plated at high densities (>1 × 10^6^ cell/cm^2^) [31]. By 1990, a methodology was introduced that allowed for the sustained culture of human epididymal cells over a period of 42 days, marking a significant advancement [32]. Epididymal epithelial cells from various species can be isolated and cultivated successfully in vitro, with some demonstrating functional capacities, particularly in promoting sperm maturation and enhancing sperm motility [33,34,35,36,37,38,39]. Despite extensive research into the relationship between epididymal epithelial cells and sperm maturation, the underlying molecular mechanisms remain poorly understood. This gap may be attributed to the inability of monolayer cultures or in vitro-passaged cells to faithfully replicate the complex biology of epididymal epithelial cells acquired in vivo, thus limiting our understanding of these intricate biological processes. Epithelial principal cell lines derived from humans [40,41], rats [42], and mice [43,44] have been employed to investigate the role of cellular communication in the epididymis [45,46,47], and to assess reproductive toxicity [47]. However, these cell lines originate from a singular source and do not fully represent the comprehensive biological functions of epididymal tissue.

Tissue culture models have successfully elucidated aspects of epididymal biology and supported drug toxicity testing [48,49,50,51,52]. Although the in vitro culture model of epididymal tissue provides a powerful platform for studying the function of the epididymis in various species [48,53], it remains operationally challenging. Previous research has demonstrated that the isolation and cultivation of primary human epididymal epithelial cells are relatively well established [54,55]. However, the in vitro culture of rodent epididymal monolayer epithelial cells has rarely been reported. Although there are a few reports, normal morphology in the culture conditions has proved difficult to maintain [56,57]. Given the diverse cellular composition of the epididymal epithelium, a 3D in vitro model provides a more comprehensive and realistic approach to investigating and understanding the intricate facets of epididymal function.

Organoids are three-dimensional structures composed of multiple cells that closely mimic the cellular structure and function of organs [58]. This similarity in structure and function facilitates the investigation of complex cell interactions and tissue development processes [59]. The emergence of organoid research has opened new avenues for both fundamental and translational research over the past decade [60]. Organoids, such as testis organoids, hold significant promise in reproductive biology and toxicology, whether in animal or human models [61,62,63]. Similarly, epididymal organoids also exhibit considerable potential. Recent in vitro studies have successfully demonstrated the formation of epididymal organoids from single-cell suspensions in different species. This review highlights recent advances in the generation of epididymal organoids and their potential applications.

## 2. The Main Function of the Epididymal Epithelium

The distribution and characteristics of the epididymal epithelium are crucial for sperm maturation, with epithelial cells exhibiting varied morphologies and functions across different regions. Sperm are generated in the seminiferous tubules of the testis. Initially devoid of motility and fertilization capabilities, spermatozoa acquire these functions during transit in the epididymal lumen—a process known as sperm maturation [9,64,65]. The components of the epididymal luminal fluid are mainly synthesized and secreted by various types of epithelial cells lining the duct [66]. The luminal fluid in the caput and corpus of the epididymis can assist sperm in acquiring motility and fertilization ability, while the luminal fluid in the cauda is beneficial for sperm storage [67]. Epididymal epithelial cells play a crucial role in establishing a highly specialized luminal microenvironment. This microenvironment is specifically tailored to promote a gradient that enhances the fertility of the sperm population contained within [68]. Despite the fact that the complexity of this process presents challenges and it lacks full comprehension, specific facets of sperm maturation have been firmly established [6,69,70].

Multiple factors are responsible for sperm maturation in the epididymal lumen. These include proteins secreted by the principal cells in the epididymis that bind to maturing spermatozoa; exosomes released by the apical plasma, called epididymal exosomes, which transport cargo to the sperm; and pH fluctuations throughout the epididymis [67,71,72,73]. The blood–epididymal barrier (BEB) is formed by apical tight junctions between the principal cells, enabling the selective transportation of molecules through the epithelium. These tight junctions consist of integral proteins that play a central role in determining the barrier’s selective permeability, thereby creating the luminal environment conducive to facilitating sperm maturation [6,74]. The maintenance of a normal epididymal epithelium is indispensable for proper sperm maturation, and epididymal dysfunction is intricately linked to infertility [40,41,75,76].

The formation of organoids in the epididymis holds great promise for revealing the underlying molecular mechanisms that regulate epididymal function. However, a crucial prerequisite for the development of epididymal organoids is the presence of stem cells or progenitor cells within the epididymis [77,78]. Are there stem cells or progenitor cells in the epididymis that can give rise to these organoids?

## 3. Basal Cell—The Prerequisite for Organoid Formation?

As early as 1925, researchers using a rat model first suggested the existence of stem cells within the epididymal epithelium [79]. Subsequent hypotheses proposed basal cells as potential stem cells in the epididymis [80,81]. Observations in unilaterally orchiectomized adult male rats revealed that basal cells exhibited a transition from an oval to a triangular and elongated shape, evolving into expanded columnar cells [79]. In vitro studies demonstrated that basal cells, identified by keratin 5 (KRT5) positivity, could differentiate into cells expressing KRT8 and connexin 26, markers typical of columnar cells [13]. These basal cells showed self-renewal and differentiation capabilities, forming organoids capable of expressing aquaporin 9 and CFTR, indicative of principal cell markers [82,83]. Furthermore, these cells secreted clusterin, a protein crucial for spermatozoa maturation [84]. Basal-cell-derived organoids exhibited self-renewal potential, maintaining newly formed organoids for at least 13 passages [84]. This evidence strongly supports the characterization of basal cells as possessing stem cell-like properties with significant self-renewal capacity [85]. Previous research has documented segment-specific gene expression and regulation within the epididymis [86,87]. Moreover, gene expression profiles of the principal cells varied between the proximal and distal segments. Interestingly, no significant differences were observed in the organoids derived from basal cells isolated from either proximal or distal epididymal regions [84]. This suggests that regional differences in gene expression may not originate solely from the specific segmental origin of basal cells.

In the epididymis, GJB2 serves as a marker for columnar cells, with its expression levels decreasing significantly as these cells differentiate into principal and other cell types [88]. GJB2 was not detected in basal cells cultured in vitro for 3 days; however, its expression became evident in cells within the acini after 7, 10, and 14 days of culture [13]. This suggests that basal cells possess the capacity to differentiate into cells resembling columnar cells. A similar mechanism has been observed in the trachea, where exposure to SO_2_ depletes ciliated cells, prompting basal cells to differentiate initially into undifferentiated progenitors. These progenitors then progress through differentiation stages to become ciliated and secretory cells, indicating a sequential two-step differentiation process [89]. Therefore, this implies the feasibility of regenerating the epididymal epithelium, potentially shedding light on the adaptability of this crucial organ in male fertility. Upon single-cell analysis, three distinct clusters of basal cells were identified, demonstrating the common expression of marker genes such as *Itga6* and *Krt14*. These clusters exhibited an enrichment of genes mainly involved in cell adhesion, membrane transport, and lipid metabolism [86]. The precise nature of these basal cell clusters—whether they represent distinct cell types, different stages of differentiation, or separate adult stem cell populations—remains ambiguous. Accordingly, the selection of KRT5-positive or ITGA6-positive cells may have biased the enrichment towards specific basal cell subpopulations [90].

On the contrary, other studies have indicated age-related characteristics in basal cells, which challenge their classification as stem cells [91]. In contrast, organs like the liver and amniotic membrane have exhibited epithelia containing expanding stem cells [92,93]. Additionally, quiescent adult stem cells with active regenerative properties have been identified in many tissues [94], such as the salivary gland [95], liver [96], intestine [97], and pituitary [98]. Therefore, although the low proliferation or expansion index of epididymal epithelial cells does not conclusively prove the existence of stem cells in the epididymis [79], ongoing debate persists regarding whether basal cells in the epididymis fulfill the criteria of adult stem cells [99]. Even without confirmation as true stem cells, basal cells likely retain differentiation potential and contribute to organoid formation [84]. Thus, despite ongoing controversy, researchers can continue to employ basal cells from the epididymis for in vitro culture and organoid studies.

## 4. Development History of Epididymal Organoids

A well-designed microenvironment in tissue and cell engineering can promote proliferation, migration, matrix production, and stem cell differentiation. Significant differences exist regarding cell–cell interactions, cellular mechanics, and nutrient access between 3D and standard 2D cell cultures, as noted by reference [100]. Nevertheless, 2D monolayer cell culture systems may not accurately simulate the observed cell development process in the in vivo physiological environment due to their inherent simplicity. This discrepancy stems from the lack of a complex, biologically rich environment. The advent of 3D cell culture approaches, which model in vivo tissue and organ interactions, has opened new avenues for studying underlying biochemical and biomechanical signals [101,102]. Given their ability to more closely mimic the in vivo environment, 3D culture systems are gaining popularity. The term “organoids” was coined in 1947 within the field of oncology [103]. With advancements in stem cell biotechnology, particularly the refinement of three-dimensional (3D) cell culture techniques, the definition of organoids has evolved to encompass 3D in vitro structures derived from pluripotent stem cells (PSCs) or adult stem cells (ASCs), exhibiting near-native microanatomy [78,104].

The first reported case of organoids used intestinal cells, which was published in *Nature* in 2007, and marked a significant breakthrough in biological research. These studies identified leucine-rich repeat-containing G-protein-coupled receptor 5 (*Lgr5*) as a specific marker gene for intestinal stem cells, enabling their characterization and purification [105]. Subsequent research revealed the capacity of adult intestinal stem cells to proliferate and differentiate both in vivo and in vitro [106,107]. These findings underscored the potential of 3D culture techniques to support ASC self-renewal and the formation of organ-like structures, thus offering promising avenues for tissue regeneration research. Since then, organoids have been successfully developed from various tissues including the stomach [108], liver [109,110], brain [111], prostate [112], mammary gland [113], testis [114,115,116], endometrium [117], fallopian tube [118,119], ovary [120], and epididymis [84,121,122,123], among others [60,124,125].

The development of 3D culture technologies has enabled the use of in vitro models to study epididymal function mechanisms [39,99,126]. Early epididymal structures have been observed to form epididymal spheroids under both 2D and 3D conditions [13,34,127,128]. This review provides an overview of the developmental history of epididymal spheroids or organoids cultivated in various species (see Figure 2). The formation process of epididymal organoids involves digesting epididymal tissue into single cells, after which the epithelial cells or basal cells within the epididymal tissue can spontaneously re-aggregate to form spheres and organoid structures, resembling the arrangement of epithelial cells in vitro [8,121]. More detailed information is summarized in Table 2.

Mou et al. first reported epididymal organoids in mice in 2016, isolating KRT5-positive basal cells to construct organoids consisting of basal and clear cells in vitro. When these basal cells were subcutaneously injected into nude mice, they differentiated and formed spherical structures comprising basal and principal cells [99]. The matrix is typically composed of the ECM (extracellular matrix), and the cell density can be set at 2 × 10^4^ cells. The basic culture medium utilizes DMEM/F12 supplemented with 25 ng/mL of EGF, and the culture cycle can extend up to 10 days. Pinel and Cyr isolated basal cells from rats and cultured epididymis-like organoids in vitro, highlighting the stem cell characteristics of basal cells. The organoids were cultured by depositing homogeneous cell suspensions as 50 μL drops onto Matrigel-coated, 24-well plates and incubating them upside down at 37 °C for 30 min to solidify the Matrigel [84]. Dufresne et al., from the same research group, utilized rat epididymis organoids to simulate epididymal development and analyze gene expression profiles through transcriptomic analysis across different stages of organoid growth [123]. These studies predominantly focused on epididymal organoids derived from caput, proximal, or distal epididymal regions, demonstrating their self-assembly and differentiation capabilities.

Based on the aforementioned studies, we extended our research to construct epididymal organoids from different regions, specifically the caput, corpus, and cauda, and analyzed their respective gene expression profiles. In our laboratory, and based on the methods used for generating human [121] and rat [84] organoids, we successfully optimized a protocol for the formation of mouse epididymis organoids, as illustrated in Figure 3. We obtained epithelial cells derived from different regions of the mouse epididymis and successfully generated organoids resembling the caput, corpus, and cauda of the epididymis in vitro. We mixed the appropriate concentration of basement membrane extract (BME) with a sufficient quantity of epididymal epithelial cells and supplemented this with EGF, testosterone, dihydrotestosterone, retinoic acid, and other additive factors. We took 10 μL of the mixed cell suspension at a low temperature and created a small droplet in a 96-well cell culture plate. Once the droplet solidified, and with the plate the right way up, we added an appropriate volume of organoid culture medium until the organoid formed. This method is much simpler than those previously reported [84].

## 5. Potential Application of Epididymal Organoids

Organoid models derived from the cells of mouse, rat, and human epididymis have been established [121,122,128]. These novel in vitro cell culture systems represent a significant advancement complementing existing epididymal cell lines and animal models. Organoids are pivotal for advancing the precision of male infertility treatment [127,129]. Similar to testicular organoids in male reproduction, epididymal organoids offer a valuable platform for the high-throughput screening of drugs and toxicity [130], although research in this area is still in its developmental stage. Limited knowledge suggests a paucity of information regarding the drug screening and clinical applications of epididymal organoids. The majority of research efforts have been directed towards establishing culture systems and studying the simulation of epididymal development. Drawing from research on organoids derived from other tissues [59,131,132,133], potential applications of epididymal organoids could include the following:

(1) Disease modeling: Epididymal organoids offer a promising avenue for disease modeling and studying various epididymal conditions such as infertility and congenital abnormalities. In males with cystic fibrosis, defects in the epididymis or vas deferens often lead to obstructive azoospermia [134]. Leir et al. utilized a human epididymal epithelial cell organoid model to elucidate the molecular mechanisms underlying male infertility mediated by CFTR in cystic fibrosis patients [121]. Thus, epididymal organoids demonstrate significant potential for studying male infertility and for screening therapeutic drugs. (2) Drug testing: Organoids cultured from prostate cancer patients have demonstrated resistance to cell growth arrest and apoptosis induced by BET inhibitors, revealing a new molecular mechanism for BET inhibitor resistance in these patients [135,136]. Recent epidemics, such as the COVID-19 pandemic, have been associated with impaired sperm in males with moderate SARS-CoV-2 infection [137,138]. The development of testis and epididymis-like organs can offer a more suitable ex vivo model for studying such acute events. Testis organoids have emerged as effective tools for organ-level reproductive toxicity screening [63,114,139], yet the role of epididymal organoids in drug screening and virus infection treatment remains unexplored. Thus, establishing epididymal models for drug screening is crucial to advance research on viral resistance in reproductive organs. (3) Reproductive biology research: Organoids can partly replicate the structural and functional characteristics of in vivo organs, facilitating the study of cell interactions and signaling pathways through organoid cultures [8,58]. Early studies indicated that epididymal epithelial cells play a role in promoting sperm maturation, offering a platform for investigating this process in vitro [34,35,37]. However, the limitations of 2D culture conditions hinder a comprehensive exploration of how the epididymal epithelium impacts sperm maturation and its underlying mechanisms [36,140,141]. Organoid technology allows researchers to manipulate cell types and observe their re-aggregation dynamics, a capability not feasible in traditional organotypic cultures [142]. At present, there is no report on whether epididymal organoids promote sperm maturation, which could be one of the directions of future research.

## 6. Conclusions

Until now, only a limited number of 3D epididymal organoids have been developed [84]. Several challenges are associated with creating 3D organoids of the epididymis. An accurate simulation of the diverse segments of the epididymis would be advantageous for studying sperm transport, maturation, acquisition of motility, and their underlying molecular mechanisms. Researchers have encountered difficulties due to the scarcity of epididymal tissue samples and the challenges involved in obtaining tissue from younger patients who do not have epididymal cancer. The advent of the male reproductive system organoids, encompassing prostatic, testicular, epididymal, and potentially seminal vesicle organoids, suggests a promising trajectory towards their integration into a cohesive, multi-organ-on-a-chip platform. Moreover, epididymal organoids serve as a valuable tool for elucidating infertility mechanisms, studying treatment efficacy, and evaluating drug toxicity. Future investigations should focus on refining them, with a critical need to elucidate the physiological and pathological contexts related to these changes. This understanding is pivotal for elucidating their implications in both clinical and physiological studies.

## Figures and Tables

**Figure 1 biomolecules-14-01026-f001:**
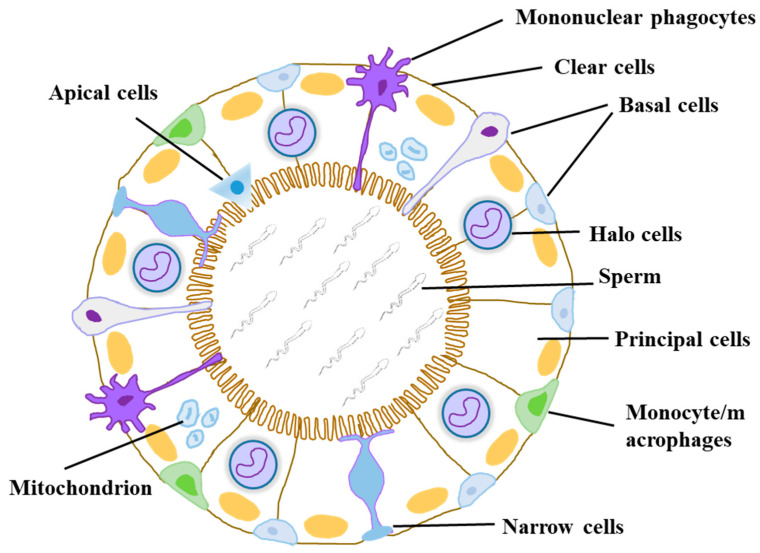
Schematic diagram of the cellular organization in a representative cross-section of the epididymis. Modified and reprinted with permission of the author (Chen et al., 2022) [24].

**Figure 2 biomolecules-14-01026-f002:**
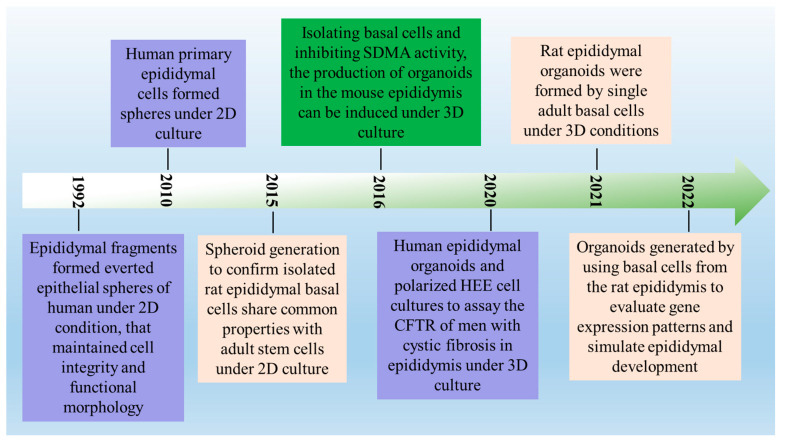
Timeline for the development of epididymal organoid cultures. A summary of key landmark studies and breakthroughs leading to the establishment of epididymal organoids in different species including human, mouse, and rat. Two-dimensional, (2D); three-dimensional, (3D).

**Figure 3 biomolecules-14-01026-f003:**
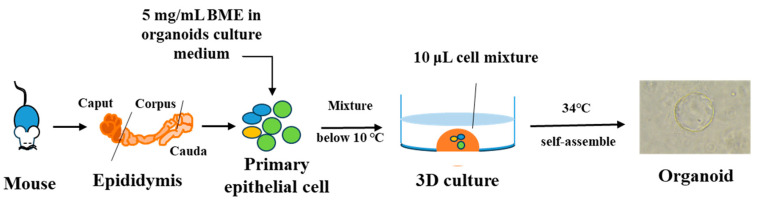
Simplified workflow for organoid formation from mouse epididymis epithelial cells in our laboratory. Epididymis from an adult mouse was sampled and enzymatically digested to obtain single-cell suspensions. The cells were cultured in extracellular matrix (basement membrane extract—BME) under 3D conditions and subsequently incubated at a temperature of 34 °C.

**Table 1 biomolecules-14-01026-t001:** Characteristics, functions, and markers of the epididymal epithelial cell.

Cell Type	Characteristic	Function	Marker	References
Principal cell	Tall, columnar shape in the proximal regions with a squared-off appearance in the distal regions, with microvilli 500 nm–1.0 mm in length and 100 nm in width forming the epididymal brush border	Secretion/Reabsorb, Merocrine, Apocrine secretions	AQP-9, CFTR, NHER1	[11,25,26]
Clear cell	An apical pole enriched with mitochondria which displays a complete and functional endocytic apparatus	Endocytic cells, proton secretion	V-ATPase, CIC-5	[27,28]
Apical cell	Present in the initial segment of the epididymis displaying a spherical nucleus at the apical pole of the epithelium	Control of inflammatory responses in the epididymis	V-ATPase, GSTM3	[11,29,30]
Basal cell	Pyramidal-shaped cells located at the base of the epithelium which directly interact with neighboring principal and clear cells through gap junctions	“Stem cell” character and “lumen-reaching” property	KRT5	[11,23]
Narrow cell	Elongated and narrow shape, present in the initial segment of the epididymis	Proton secretion and acidification of the epididymal fluid	V-ATPase, CIC-5	[27,28]

**Table 2 biomolecules-14-01026-t002:** Timeline for the development of epididymal organoids in different species.

Year	Species	Results	References
1992	Human	Epididymal fragments formed everted epithelial spheres that maintained cell integrity for 5–7 days.	[34]
2010	Human	Epididymal cells formed spheres for at least 20 days.	[128]
2015	Rat	Basal cells may represent an epididymal stem cell population.	[13]
2016	Mouse	Expanded epididymis basal cells efficiently generated organoids in Matrigel.	[99]
2020	Human	Epididymal cells generated organoid and provided the tool for studying cystic fibrosis (CF) in infertile men.	[121]
2021–2022	Rat	Basal cells generated organoids capable of secreting function and columnar cells represent an epididymal stem/progenitor cell population.	[84,123]

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
