# Peer review of "Advancement and Potential Applications of Epididymal Organoids"

_biomolecules, 2024, doi:10.3390/biom14081026_

Round 1

Reviewer 1 Report

Comments and Suggestions for Authors

In this Manuscript the authors review epididymal organoids, and how they may be used to study the process of mammalian sperm maturation. This is an interesting topic although there is a lot o room for improvement, as the paper is very poorly organized/written, with many repetitive sentences on in several (not all) sections.

The Introduction is very confusing, with several aspects mixes together. In fact this is not needed in a review (the Abstract is enough). The review  should start with the structure of the organ, its different parts, the different cells the compose it and their roles (moving Figure 1 up). A Table with cell characteristics, functions and markers would also be useful.

After that go into more detail, such as infertility. This part on epididimus-related male infertility is also very poorly organized and seems as if ideas were joined together with no logical sequence/structure.

Then there should be a general section on organoids, only then going to epididymus organoids.

-Figure 2 is potentially useful but must be complemented with a Table showing the different papers that serve as milestones in this research. What models were used in which reference and what was achieved in what species when.Figure 2 lacks all this information, unlike what the authors state in the text.

Throughout the paper the species used to collect different information are not always mentioned, and this should always be the case.

Sponge aggregation is not akin to organoid technology, it is a regeneration strategy, an organoid is not the functional organ. This comparison is farfetched.

Also in line 302 the authors state: “can reproduce the original structure and function of in vivo organs”. This is blatantly untrue, no testicular organoid has formed sperm de novo in permanence, for example,  not epididymus organoids have given motility to testicular sperm; brain organoids do not exactly recapitulate thought, etc. There should be much more care with this type of fantastic statement. Organoids are very relevant models, they are not magical. Often the authors contradict themselves: organoids cannot do almost everything in one part of the paper and yet so much is still left to be done in another sentence...

Lines 247-248 The authors should explain here what they analyzed and how organoids are truly more physiological than 2D models.

The beginning of Part 4 is again very repetitive, and should be edited, many issues have been noted previously and the paper is not that long.

In that regard Part 4 and 5 can be joined, the Conclusion and Prospects (not Prospect)  section adds nothing to the paper as currently written, it just repeats what the authors mostly said in the previous section.

Comments on the Quality of English Language

The English is often not grammatically correct and needs to be improved in many places. There are also clear mistakes and places where the authors categorically state what they cannot state in such a manner. The paper must be thoroughly revised in this aspect.

Author Response

Comment 1: In this Manuscript the authors review epididymal organoids, and how they may be used to study the process of mammalian sperm maturation. This is an interesting topic although there is a lot of room for improvement, as the paper is very poorly organized/written, with many repetitive sentences on in several (not all) sections.

Response: Thank you for your valuable comments on improving our manuscript. We have thoroughly reviewed the paper, removed all repetitive sentences, and reorganized the content accordingly.

Comment 2: The Introduction is very confusing, with several aspects mixes together. In fact, this is not needed in a review (the Abstract is enough). The review should start with the structure of the organ, its different parts, the different cells the compose it and their roles (moving Figure 1 up). A Table with cell characteristics, functions and markers would also be useful.

Response: Thank you for your feedback. We have reorganized the manuscript and included a new table (Table 1) that outlines cell characteristics, functions, and markers in the introduction. We appreciate your efforts in improving our article.

Comment 3: After that go into more detail, such as infertility. This part on epididimus-related male infertility is also very poorly organized and seems as if ideas were joined together with no logical sequence/structure.

Response: Thank you for your efforts in improving our article. We have sought professional assistance to refine this section by enhancing sentence structure and enriching its content.

Comment 4: Then there should be a general section on organoids, only then going to epididymis organoids.

Response: Thank you for your comments. In the revised manuscript, we have added a paragraph on organoids prior to the section on epididymis organoids (lines 178-188).

Comment 5: -Figure 2 is potentially useful but must be complemented with a Table showing the different papers that serve as milestones in this research. What models were used in which reference and what was achieved in what species when Figure 2 lacks all this information, unlike what the authors state in the text.

Response: Thank you for your careful work. We have added Table 2, which provides more information related to Figure 2, in order to enhance its completeness.

Comment 6: Throughout the paper the species used to collect different information are not always mentioned, and this should always be the case.

Response: Thank you for your comments. We have thoroughly reviewed and rearranged the paper.

Comment 7: Sponge aggregation is not akin to organoid technology, it is a regeneration strategy, an organoid is not the functional organ. This comparison is farfetched.

Response: We have removed the comparison between sponge aggregation and organoid, as it was suggested that this comparison is far-fetched. Thank you for your efforts to improve our article.

Comment 8: Also in line 302 the authors state: “can reproduce the original structure and function of in vivo organs”. This is blatantly untrue, no testicular organoid has formed sperm de novo in permanence, for example, not epididymus organoids have given motility to testicular sperm; brain organoids do not exactly recapitulate thought, etc. There should be much more care with this type of fantastic statement. Organoids are very relevant models, they are not magical. Often the authors contradict themselves: organoids cannot do almost everything in one part of the paper and yet so much is still left to be done in another sentence...

Response: Thank you for your suggestion. We have reviewed our description and modified the sentence “organoids can reproduce the original structure and function of in vivo organs” to “Organoids can partly replicate the structural and functional characteristics of in vivo organs.” We apologize for our earlier less rigorous expression.

Comment 9: Lines 247-248 The authors should explain here what they analyzed and how organoids are truly more physiological than 2D models.

Response: Thank you for your efforts to improve our article. We apologize for any confusion caused by our previous descriptions, and we have removed the text in lines 247-248. In response to the insightful feedback provided by the esteemed reviewers regarding the comparison between organoids and two-dimensional (2D) models, it is essential to clarify the specific details of our analysis. While two-dimensional (2D) cell cultures are widely employed in many biomedical studies, they are often regarded as offering only simple physical interactions between cells, lacking the structural complexity of tissues. In contrast, organoids present several advantages over 2D cell cultures, including a closer approximation to physiological cellular compositions and behaviors, greater genomic stability, suitability for biological transfection, and compatibility with high-throughput screening. Moreover, organoid models are generally easier to manipulate than animal models, making them valuable for studying the mechanisms underlying disease initiation and progression.

Comment 10: The beginning of Part 4 is again very repetitive, and should be edited, many issues have been noted previously and the paper is not that long.

Response: Thank you for your efforts to improve our article, we have deleted the beginning of Part 4 and reorganized the sentences and paragraph to enhance readability.

Comment 11: In that regard Part 4 and 5 can be joined, the Conclusion and Prospects (not Prospect) section adds nothing to the paper as currently written, it just repeats what the authors mostly said in the previous section.

Response: Thank you for your valuable feedback. We have combined sections 4 and 5 and removed the Conclusion and Prospects. Additionally, we have reorganized the sentences and paragraphs to enhance readability.

Reviewer 2 Report

Comments and Suggestions for Authors

This is a really interesting and innovative review related to the advancement and potential applications of epididymal organoids. Of course, the subject is extremely relevant not only for human but also for animal reproduction, specially related to the comprehension of epididymal physiology, with a consequent application for the infertility from epididymal origin. I have only few comments as listed below.

In general, manuscript is really well written and organized. The topics bring us recent advances related to the subject and the sentences are well connected.

Authors first stated the composition of epididymal epithelium and the characteristics of the basal cells (At this point, I suggest to change "characters" for "characteristics", in the tittle of Topic 2). This topic is well written, and authors largely detailed the physiology of blood-epididymal barrier as well as the functions of each cell that compose epididymus. 

Then, authors explained the historical steps for the development of epididymal organoids in details. Despite this, I suggest authors to include some more details related to 3D system used for this purpose. What kind of matrix was used in the different studies and which is the best one? How long last the culture for producing adequate organoids? Which is the media? Which are the most used supplements to the media? A final question: What enzymes are used for epididymal cells digestion? How are the epididymal cells selected for using in this kind of culture? How is the efficiency of the culture analyzed?

I have no more comments on the further topics. Applications and prospectes are well written and discussed. Moreover, Illustrations were well designed and are very informative throughout the manuscript. 

Author Response

This is a really interesting and innovative review related to the advancement and potential applications of epididymal organoids. Of course, the subject is extremely relevant not only for human but also for animal reproduction

n, specially related to the comprehension of epididymal physiology, with a consequent application for the infertility from epididymal origin. I have only few comments as listed below.

Comment 1: In general, manuscript is really well written and organized. The topics bring us recent advances related to the subject and the sentences are well connected.

Response: Thank you for your affirming of our work.

Comment 2: Authors first stated the composition of epididymal epithelium and the characteristics of the basal cells (At this point, I suggest to change "characters" for "characteristics", in the tittle of Topic 2). This topic is well written, and authors largely detailed the physiology of blood-epididymal barrier as well as the functions of each cell that compose epididymus.

Response: Thank you for your careful job, we have removed the term "characters" to better align with the content requirements. Additionally, we have included a table that provides more comprehensive information regarding the characteristics, functions, and markers of the epididymal epithelium. We appreciate your affirmation of our efforts.

Comment 3: Then, authors explained the historical steps for the development of epididymal organoids in details. Despite this, I suggest authors to include some more details related to 3D system used for this purpose. What kind of matrix was used in the different studies and which is the best one? How long last the culture for producing adequate organoids? Which is the media? Which are the most used supplements to the media? A final question: What enzymes are used for epididymal cells digestion? How are the epididymal cells selected for using in this kind of culture? How is the efficiency of the culture analyzed?

Response: The matrix, which includes the extracellular matrix (ECM) and bioengineered extracellular matrix (BEM), supports the cultivation of epididymal organoids for periods ranging from several days to months. The cultivation medium for most epididymal organoids typically comprises DMEM/F12 as the base liquid, supplemented with specific growth factors, such as epidermal growth factor (EGF) and BMP4 inhibitors. The separation of epididymal cells is generally facilitated by collagenase I or IV. Given that the epididymal epithelium constitutes the majority of the epididymis, the primary cells obtained from this separation can effectively be utilized for the formation of epididymal organoids in a controlled environment. In general, the efficiency of organoid formation is lower for epididymal organoids; however, the formation efficiency of organoids derived from separated basal cells is comparatively higher. Detailed information on these processes is discussed in the main text.

Comment 4: I have no more comments on the further topics. Applications and prospectes are well written and discussed. Moreover, Illustrations were well designed and are very informative throughout the manuscript.

Response: Thank you for your affirming of our work.

Reviewer 3 Report

Comments and Suggestions for Authors

Comments to the manuscript: biomolecules 3115234

This review describes the advancement and some applications of Epididymal Organoids. This topic is very important.

This review appears to be well conducted, but it needs revision.

The following comments only represents personal opinions

General Comments and questions

The written quality of this Manuscript needs to be improved, since it contains

spelling, grammar and wording mistakes. In addition, it must be revised by a native English speaker or professional editing service.

Specific comments: 

What is the novelty of this paper.

Could you please add a paragraph about the application of cell culture. 

Comments on the Quality of English Language

Moderate editing of English language required

Author Response

This review describes the advancement and some applications of Epididymal Organoids. This topic is very important. This review appears to be well conducted, but it needs revision. The following comments only represents personal opinions

Comment 1 The written quality of this Manuscript needs to be improved, since it contains spelling, grammar and wording mistakes. In addition, it must be revised by a native English speaker or professional editing service.

Response: Thank you for your advice, we have polished the article by a professional agency or individual.

Comment 2: What is the novelty of this paper. Could you please add a paragraph about the application of cell culture.

Response: This review aims to provide a reference for organoid generation in the epididymis. The novelty of this paper is threefold. Firstly, it offers a comprehensive examination of recent research on epididymal epithelial cells, including their functions and characteristics. Secondly, we elucidate the relationship between basal cells and epididymal organoids, detailing the developmental history of these organoids, which has not been previously reported. Moreover, we present an optimized scheme for the production of epididymal organoids that is simpler than previously described methods. The article also explores potential applications of epididymal organoids, thereby establishing a direction for future research in this field. Additionally, we have included a paragraph discussing the application of cell culture in the introduction.

Round 2

Reviewer 1 Report

Comments and Suggestions for Authors

The authors have addressed my concerns. I have no further comments.